# The Need for Artificial Intelligence Based Risk Factor Analysis for Age-Related Macular Degeneration: A Review

**DOI:** 10.3390/diagnostics13010130

**Published:** 2022-12-30

**Authors:** Abhishek Vyas, Sundaresan Raman, Janani Surya, Sagnik Sen, Rajiv Raman

**Affiliations:** 1Birla Institute of Technology & Science, Pilani 333031, India; 2Sankara Nethralaya Medical Research Foundation, Chennai 600006, India; 3Moorfields Eye Hospital, London EC1V 2PD, UK; 4Aravind Eye Hospital, Madurai 625020, India

**Keywords:** age-related macular degeneration, artificial intelligence, statistical techniques, machine learning, deep learning, identifying risk factors, personalized care

## Abstract

In epidemiology, a risk factor is a variable associated with increased disease risk. Understanding the role of risk factors is significant for developing a strategy to improve global health. There is strong evidence that risk factors like smoking, alcohol consumption, previous cataract surgery, age, high-density lipoprotein (HDL) cholesterol, BMI, female gender, and focal hyper-pigmentation are independently associated with age-related macular degeneration (AMD). Currently, in the literature, statistical techniques like logistic regression, multivariable logistic regression, etc., are being used to identify AMD risk factors by employing numerical/categorical data. However, artificial intelligence (AI) techniques have not been used so far in the literature for identifying risk factors for AMD. On the other hand, artificial intelligence (AI) based tools can anticipate when a person is at risk of developing chronic diseases like cancer, dementia, asthma, etc., in providing personalized care. AI-based techniques can employ numerical/categorical and/or image data thus resulting in multimodal data analysis, which provides the need for AI-based tools to be used for risk factor analysis in ophthalmology. This review summarizes the statistical techniques used to identify various risk factors and the higher benefits that AI techniques provide for AMD-related disease prediction. Additional studies are required to review different techniques for risk factor identification for other ophthalmic diseases like glaucoma, diabetic macular edema, retinopathy of prematurity, cataract, and diabetic retinopathy.

## 1. Introduction

Diabetes has become the fifth leading cause of blindness across the globe, due to which ophthalmic diseases are a global concern. Retinal disorders can cause blindness, but early diagnosis and timely treatment can prevent vision loss. Therefore, there is a dire need for automated diagnosis systems to assist in the early diagnosis of ophthalmic disease. There exist many retinal diseases. Cataract refers to the clouding of the lens, and in glaucoma the optic nerve of the eye that provides information to the brain is damaged, which can lead to gradual vision loss when left untreated and in diabetic retinopathy, where the blood vessels of the eye are damaged owing to the complication of diabetes. Researchers developed automatic detection systems to detect cataract [1,2,3,4], glaucoma [5], and diabetic retinopathy [6,7].

Age-related macular degeneration is also a leading cause of visual impairment and severe vision loss and is the most common form of maculopathy leading to vision loss in people [8,9]. AMD affects individuals over the age of 55 years. According to the world report on vision, of the estimated 196 million people suffering globally from age-related macular degeneration, 10.4 million (5.3%) have moderate or severe distance vision impairment or blindness from more severe forms of the condition [10].

A meta-analysis [11] suggested that the number of people in Europe affected by any AMD is expected to increase by 15% by 2050. AMD can be divided into subgroups according to the presence/absence of neo-vascularization–dry/non-neo-vascular and wet/neovascular AMD (nAMD). Several risk factors of AMD have been identified: age; gender; race; obesity; hypertension; smoking; sunlight; diet; phenotypic; demographic; environmental; genetic; alcohol consumption; and molecular risk factors; etc. [12,13,14,15,16,17,18,19,20,21,22,23,24,25]. Therefore, using risk factors as biomarkers for predicting AMD and AMD progression is a significant area of research

Artificial intelligence (AI) in diagnosis and prognosis represents a paradigm shift in healthcare. Artificial intelligence (AI) has already demonstrated proof-of-concept in medical fields such as radiology, pathology, and dermatology, which are similar to ophthalmology as they are deeply rooted in diagnostic imaging, which is the most prominent application of AI in healthcare. AI in medicine has overwhelming advantages: it can detect and learn features from large volumes of imaging data using efficient algorithms, assisting clinical practice. It can foster personalized medicine and help reduce diagnostic and therapeutic errors. In addition, AI can correlate novel features and recognize disease-specific patterns to gain innovative scientific insight. AI aims to contribute to better care outcomes and improve the productivity and efficiency of care delivery.

Many statistical methods have been designed to identify important risk factors from clinical data in the context of risk factors. However, an AI-based risk identification tool is needed to identify risk factors for ophthalmic diseases. This will nullify practitioner-dependent biases seen commonly in assessments of patients and hence the conventional methods of identifying risk factors. Novel treatments are being widely investigated in several clinical trials for both forms of AMD [26,27]. A meta-analysis has shown promising results for the diagnostic accuracy of the machine learning classifiers for AMD and its implementation in clinical settings [28]. Hence, using AI with deep learning tools has excellent potential in AMD, for diagnostic purposes–while allowing for a more efficient and accurate approach–to prognostication of affected individuals and perhaps to directly determine the efficacy of investigational medical products. 

This review aims to summarize the literature related to techniques used to study risk factors of AMD and investigate AI-based options available for the same. 

The paper is organized as follows: Section 2 presents the methods used in this research. Section 3 reviews statistical techniques used to identify risk factors for AMD. Section 4 provides a review of artificial intelligence techniques used in AMD diagnosis. Section 5 discusses the significance of AI over traditional statistical methods. Section 6 includes the discussion, and Section 7 lists the conclusion. 

## 2. Methods

### 2.1. Study Selection

#### Search Terms

We used PubMed as our primary electronic search engine for looking into published articles related to AMD from 1991 to 2019. The search terms used were “age-related macular degeneration,” “risk factors,” “prevalence,” and “incidence.” The search strategy used both text word searches and subject headings. Additionally, initial search terms were updated after searching the reference lists of relevant articles. The articles were restricted to only the English language.

### 2.2. Inclusion and Exclusion Criteria

After preliminary searches, criteria were developed in an iterative process. We included studies on AMD, which identified risk factors using different statistical techniques. All risk factors studied, from clinical variables to genes, were considered for inclusion. Studies that did not mention statistical techniques to identify the risk factors were excluded. 

### 2.3. Selecting Studies

A total of 118 articles were retrieved from PubMed after the initial search, of which 34 articles met this study’s objective and were selected for review. The flow diagram describing the study selection is depicted in Figure 1. The year-wise classification of included articles is shown in Figure 2.

## 3. Statistical Techniques for Risk Factor Identification

In medical research, statistical analyses are an essential component that can further the understanding of risk factors, treatment effects, and other aspects of the disease when appropriately applied. Statistics has become an integral part of research in ophthalmology, and its use to evaluate experimental data in ophthalmology has increased. Figure 3 depicts the various application of statistical methods from a clinician’s perspective.

Table 1 illustrates the studies included in this review, the overview of the risk factors identified, and the statistical tools used to identify them. The commonly identified risk factors identified may be divided into ocular-based factors (focal hyperpigmentation, drusen, slow choroidal filling, cataract, hyperopia), susceptibility-based risk factors (age, gender, race, hypertension, cardiovascular status, body mass index, obesity, and cholesterol), exposure-related risk factors (smoking, alcohol, physical activity) and genetics (Table 1). Figure 4 depicts the classification by percentage of techniques used in included articles in Table 1.

The review identified 34 prospective studies investigating risk factors for AMD. There was good evidence that risk factors like smoking, alcohol consumption, previous cataract surgery, age, high-density lipoprotein (HDL) cholesterol, BMI, female gender, and focal hyper-pigmentation were more often associated with being independent risk factors for AMD. One study showed a significant association between cardiovascular disease and the risk factors associated with AMD. In contrast, two studies showed no significant association between cardiovascular disease and its risk factors with AMD. Therefore, more investigation is needed to identify the association of cardiovascular disease and its risk factors with AMD.

## 4. Artificial Intelligence in AMD

In AMD pathogenesis, genetics plays a critical role. Many variants associated with AMD have been identified by sequencing studies and genome-wide association studies [63]. Figure 5 depicts the various applications of artificial intelligence from a clinician’s perspective. The input to AI can be data of the following types: numerical/categorical, fundus images, and optical coherence tomography (OCT) volumes. The most common imaging modalities in AI for AMD are OCT, color fundus image, and fundus autofluorescence (FAF).

Based on the input to the AI techniques, the application of AI can be divided into the following categories:

### 4.1. Lesion Detection, Quantification, and Extraction

A European study was done by Grinsven et al. [64] to detect and quantify drusen on color fundus photographs in 407 eyes without AMD or with early to moderate AMD. This study demonstrated that for detecting the presence of drusen and estimating the area, it achieved an intraclass correlation coefficient (ICC) larger than 0.85 and was in keeping with experienced human observers. Consequently, another algorithm was explored for the automatic detection of reticular pseudo-drusen (RPD) [65]. Automated RPD quantification achieved an ICC of 0.7, similar to the observers. Consequently, Liefers et al. [66] used a deep learning model to segment and quantify retinal features in individuals with atrophic AMD and nAMD. The mean ICC obtained was 0.66 ± 0.22 and 0.62 ± 0.21 for the model and observers, respectively. 

### 4.2. Automated Image Segmentation

Schmidt-Erfuth et al. [67] analyzed OCT volume scans features–intraretinal cystoid fluid (IR), subretinal fluid (SRF), and pigment epithelial detachments (PED) to evaluate the predictive potential of machine learning in terms of best-corrected visual acuity (BCVA). A modest correlation was found between BCVA and OCT at baseline (R^2^ = 0.21). Subsequently, the same group used a deep learning method and a convolutional neural network (CNN) to accurately measure fluid response to anti-vascular endothelial growth factor (VEGF) treatment in neovascular AMD [68] in the HARBOR study. For this purpose, the group used automatic volumetric quantification data of fluid volumes in the OCT. Subsequently, the authors also validated the retinal fluid volumes (intraretinal fluid (IRF), subretinal fluid (SRF), and pigment epithelial detachment (PED)) as important biomarkers in neovascular AMD [69]. Lee et al. [70] utilized a deep learning framework to perform automated diagnosis and segmentation of retinal diseases. They created a key OCT image segmentation model. The authors applied this methodology in 14,884 clinically heterogenous scans.

### 4.3. AMD Classification

Yim et al. [71] combined 3D OCT images and automatic tissue maps in individuals with nAMD in one eye to predict progression in the contralateral eye to nAMD. This system outperformed five out of six experts and achieved a sensitivity of 80% at 55% specificity and 34% specificity at 90% sensitivity. Yan et al. [72] used data of disease severity phenotypes and fundus images available at baseline and follow-up visits over 12 years from AREDS to predict late AMD progression. They achieved an average AUC value of 0.85 when fundus images were coupled with genotypes to predict late AMD progression. Only fundus images showed a middle area under the ROC curve value of 0.81. Peng et al. [73] combined deep learning (DL) and survival analysis to develop, train, and validate a framework for predicting individual risk of late AMD. The model achieved a 5-year C-statistic of 86.4 when validated against an independent test data set of 601 participants, which substantially exceeded that of retinal specialists using two existing clinical standards of 81.3 and 82.0, respectively. Ajana et al. [74] used genotypic, lifestyle, and phenotypic factors to develop a prediction model for advanced AMD. The training data set included Rotterdam Study I [75] (RS-I) enrolled participants. The validation dataset included antioxidants, lipides essentiels, nutrition et maladies oculaires [76] (ALIENOR) study enrolled participants. The cross-validated AUC estimation in RS-I was 0.92 at five years, 0.92 at ten years, and 0.91 at 15 years. In ALIENOR, the AUC reached 0.92 at five years. Seddon et al. [77] calculated the AMD progression risk score to discriminate progressors from nonprogressors to advanced AMD based on demographic, ocular, behavioral, treatment, and genetic factors. They obtained a C-statistic score of 0.83, compared to C statistics for coronary heart disease (CHD), 0.79 for white men, and 0.83 for white women in the Framingham study cohort, and somewhat lower in several replication samples [78].

Seddon et al. [79] included time-varying progression rates up to 12 years, macular drusen size in both eyes at baseline, AMD status at baseline, six genetic variants, and environmental and demographic factors to build a model for AMD progression. The model’s AUC for progression at ten years with drusen size, environmental covariates, and genetic factors was 0.915 in the total sample. Klein et al. [80] constructed a risk assessment model to develop advanced AMD incorporating phenotypic, demographic, environmental, and genetic risk factors. The model did well on performance measures, with excellent discrimination (C statistic = 0.872) and excellent calibration and overall performance (Brier score at five years = 0.08). Seddon et al. [81] developed an online application and a predictive model. The online application assists in clinical decision-making and is available at www.seddonamdriskscore.org. The model included age, ten genetic loci, sex, BMI, education, baseline AMD status, and smoking, and the AUC for progression to advanced AMD over ten years was 91.1%. Spencer et al. [82] combined the results from the grammatical evolution of neural networks (GENN) and logistic regression models using a consensus approach to build an algorithm using a constellation of environmental risk factors and knowledge of each individual’s particular genetic profile, which was successful in differentiating between low and high-risk groups for AMD with a sensitivity of 77.0% and specificity of 74.1%.

Fraccaro et al. [83] used black-box methods, such as random forests, AdaBoost, and SVM, as well as white-box techniques, including decision trees and logistic regression, to develop models to diagnose AMD, including demographics, depigmentation area, and, for each eye, presence/absence of significant AMD-related clinical signs (retinal pigment epithelium, soft drusen, defects/pigment mottling, subretinal fluid, subretinal hemorrhage, macula thickness, subretinal fibrosis, macular scar). The model’s logistic regression, AdaBoost, and random forests achieved an AUC of 0.92, followed by decision trees and SVM with an AUC of 0.90. Shin et al. [84] used ocular and systemic factors to develop a risk prediction model for the progression of AMD in Koreans. The model achieved a C statistic of 0.84, indicating excellent predictive power. The fundus images were used for AMD grading; they can also be used with genotypes to predict the probability of late AMD progression. Such predictions can slow the disease progression by urging the patients to start preventative care beforehand since late AMD is irreversible. 

The review of AI techniques for AMD as described in Section 4.1, Section 4.2 and Section 4.3 is summarized in Table 2.

## 5. Significance of AI over Traditional Statistical Methods

### Open Problems

Statistical methods can work only with numerical or categorical data. In contrast, artificial intelligence (AI) can detect AMD automatically. AI techniques can assist in extracting the vascular skeleton and thus compute features like tortuosity, fractal index, thickness, and vessel density of blood vessels in a fundus image. AI can also detect and quantify drusen present in a fundus image. Moreover, automatic image segmentation can also be performed using AI. 

Traditional statistical methods rely on strong assumptions, such as the additivity of the parameters within the linear predictor, the type of error distribution, and proportional hazards. These assumptions are often overlooked in the scientific literature and are not met in clinical practice. For instance, when studying survival in gastric cancer patients, the assumption of proportional hazards has been violated, as nodal status and the prognostic significance of the depth of tumor invasion tend to decrease with increasing follow-up. At the same time, the loss of the TP53 gene and the histology acquire prognostic importance after at least two years of follow-up [85]. On the other hand, machine learning (ML) techniques in AI have considerable flexibility and are free from a priori assumptions. 

Traditional statistical approaches often fail because they make an a priori selection of the variables to be considered. For instance, a Cochrane review in gastric cancer surgery dealing with the extension of lymphadenectomy was later withdrawn and criticized because it failed to assess the quality of surgical procedures under comparison [86]. Whereas in ML any number of features can be chosen based on all the available information. 

Traditional regression models show several limitations in choosing the most important risk factors when there are many predictors and few observations, such as in transcriptomics, genomics, metabolomics, and proteomics [87]. In contrast, ML is particularly suited for such situations. Therefore, it is possible to use numerous approaches to apply small datasets in building ML predictive models. 

Traditional statistical methods can only address interactions between single potential confounders and the primary determinant. For instance, in gastric cancer patients, the effect of the surgical approach on survival is modulated by histology and tumor stage [88]. However, within a Cox model, it is not easy to highlight this second-order interaction [89], as the interaction between lymphadenectomy and histology becomes apparent after the first two years of follow-up. ML can also efficiently address such interactions. Furthermore, ML algorithms can analyze various data types (imaging data, laboratory findings, and demographic data) and integrate them into predictions for illness risk, prognosis, diagnosis, and appropriate treatments [90].

## 6. Discussion

Out of the included studies, the review found that for identifying risk factors, 32.35% used logistic regression, 8.82% of each used univariate & multivariate analyses, multivariable logistic regression, multivariate stepwise logistic regression, generalized estimating equation logistic regressions, 5.88% used polychotomous logistic regression and 2.94% of each used Poisson regression, unconditional logistic analysis, standard bivariate, and multivariate analyses, multivariate Cox regression, logistic regression, and Mantel-Haenszel analysis, survival analysis and Cox proportional hazards regression. The classification by number and percentage of techniques used in included articles is depicted in Figure 3. 

AI techniques often used are logistic regression and deep learning for predicting AMD. The metrics evaluated by the AI techniques are not comparable due to the different datasets used in these studies. The prediction of AMD can be done by acquiring dataset of OCT volumes, color fundus images, and clinical data of risk factors. However, there is a tradeoff between the cost of obtaining the dataset and the metrics (accuracy, AUC, etc.) of the AI models received to predict AMD. In this context, the cost of obtaining OCT volumes is higher than obtaining color fundus images which is higher than obtaining clinical risk factor data. In this context, Yim et al. [71] demonstrated that an AI system using deep learning which combined 3D OCT images and automatic tissue maps in individuals with nAMD in one eye to predict progression in the contralateral eye to nAMD. This system outperformed five out of six experts and achieved a sensitivity of 80% at 55% specificity and 34% specificity at 90% sensitivity. Grinsven et al. [64] developed a supervised learning algorithm to detect and quantify drusen on color fundus photographs without AMD or with early to moderate AMD. The system achieved areas under the receiver operating characteristic (ROC) curve of 0.948 and 0.954 for automatic AMD risk assessment, which was similar in performance compared to human observers. Moreover, Ajana et al. [74] used genotypic, lifestyle, and phenotypic factors to develop a prediction model for advanced AMD and achieved an AUC achieved of 0.92. Therefore, using OCT volumes, fundus images, and clinical data result in similar performance if the metrics are compared to predict presence of AMD. If a method performs only moderately better using OCT volumes of data as compared to using fundus or clinical data, then the method may not prove economical. Therefore, there is always a tradeoff between the cost of obtaining the data and the metrics achieved by the AI models to predict AMD.

AI-based methods can be vital in identifying potential biomarkers for guiding targeted therapy in ophthalmology. Many risk factors are embedded in the high dimensional data produced by various imaging modalities. AI can process this high-dimensional data to find some risk factors for AMD, whereas statistical methods do not have high-dimensional data as input. Saha et al. [91] used deep learning for the automated identification of these OCT-based AMD biomarkers.

Despite the AI-based models showing a high level of accuracy in many of the diseases in ophthalmology, there are still many technical and clinical challenges for real-time deployment and clinical implementation of these models in clinical practice. These challenges could arise in both the research and clinical settings. Many studies’ training datasets are from relatively homogeneous populations [92,93,94]. AI training and testing using retinal images are subject to numerous variabilities, including the field of view, the width of the field, image quality, image magnification, and participant ethnicities. Diversifying the data set regarding image-capture hardware and races could help address this challenge [95]. 

The limited availability of large amounts of data is another challenge in developing AI models in ophthalmology. The software will likely produce inaccurate outcomes if the training set of images given to the AI tool is too small or not representative of natural patient populations. In addition, more evidence on obtaining high-quality ground-truth labels is required for different imaging tools. 

Many DL systems in AI have reported a robust diagnostic performance, although some papers did not show how the power calculation was performed for the independent data sets. A power calculation should consider the following: the prevalence of the disease, type 1 and 2 errors, CIs (confidence intervals), and desired precision. The desired operating threshold should be first preset on the training set, followed by an analysis of performance metrics such as sensitivity and specificity on the test set to assess the calibration of the algorithm.

AI is adopted in healthcare, but it is still not on the horizon as clinicians and patients are concerned about AI and DL being ‘black boxes.’ It is not only the quantitative algorithmic performance in healthcare but the underlying features essential to improve physicians’ acceptance through which the algorithm classifies the disease. Generating heat maps highlighting the regions of influence on the image which contributed to the algorithmic conclusion may be the first step, although such maps are often challenging to interpret [96]. Explainable artificial intelligence (XAI) can also be used, a set of methods and processes that allows human users to comprehend and trust the results and output created by machine learning algorithms.

There are some limitations to the review. First, the review is limited to a certain number of studies and the associated risk factors. Second, the review is limited to only AMD. It can be further extended to other ophthalmic diseases like glaucoma, retinopathy of prematurity, cataract, diabetic retinopathy, etc. 

## 7. Conclusions

To the best of our knowledge, this review is the first of its kind which analyzes the techniques for identifying the risk factors and predicting the disease using risk factor datasets in ophthalmology. Other reviews in the literature find modifiable and non-modifiable risk factors related to AMD. However, such a review of techniques to identify risk factors for AMD is so far not looked upon. In the study, logistic regression was found as the most used technique to identify risk factors for AMD. This review has highlighted that ML techniques can also be used for similar purposes. The study has demonstrated that statistical methods were used to determine the risk factors for AMD to a large extent. AI-based tools have already started managing epidemics and discovering potential drugs. Therefore, AI technology is more suitable to play a significant role in identifying risk factors in ophthalmology. To a reasonable extent, AI techniques play an essential role in predicting AMD. They have great potential to be used for personalized care in diagnosing, prognosis, and treating diseases in ophthalmology. Future studies can focus on novel analysis methods and biomarkers diagnosing AMD. For example, many patients with diabetic retinopathy do not respond well to current therapeutics. Therefore, new analytical techniques related to molecular biomarkers should accelerate progress in recent research [97]. In this context, erythrocyte membrane fluidity has been found as a biomarker for diabetic retinopathy [98]. Novel segmentation methods should be developed to unveil metabolic features as future work in this research [99,100]. 

## Figures and Tables

**Figure 1 diagnostics-13-00130-f001:**
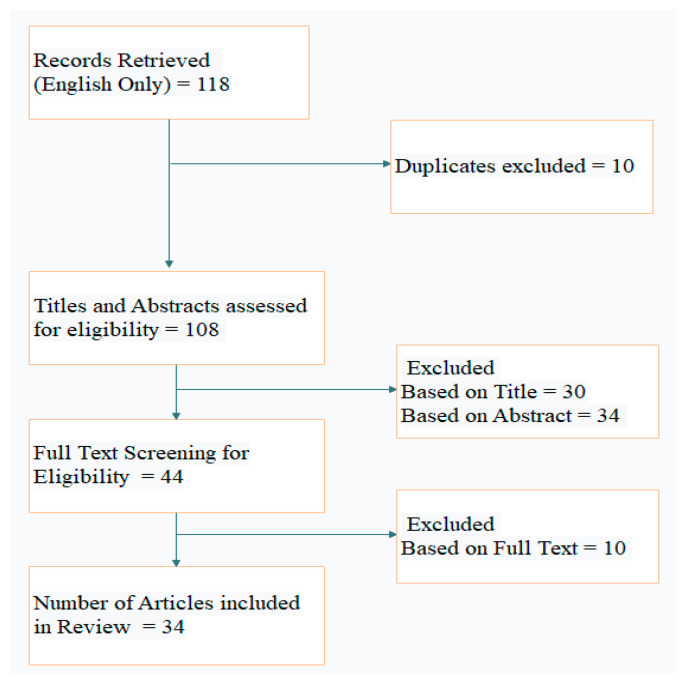
Flow diagram describing study selection.

**Figure 2 diagnostics-13-00130-f002:**
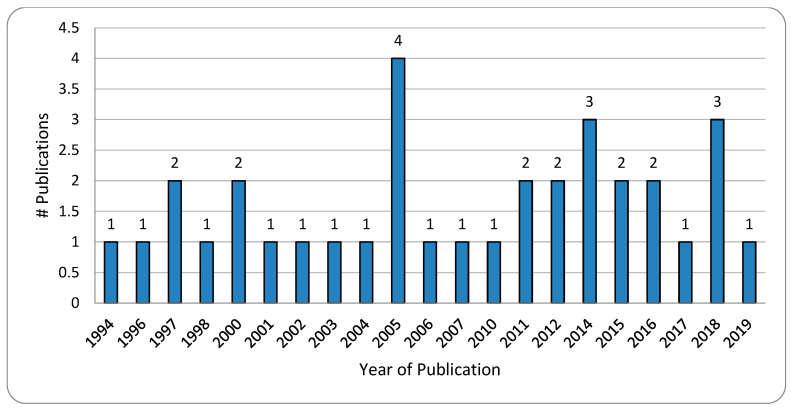
Year-wise classification of included articles.

**Figure 3 diagnostics-13-00130-f003:**
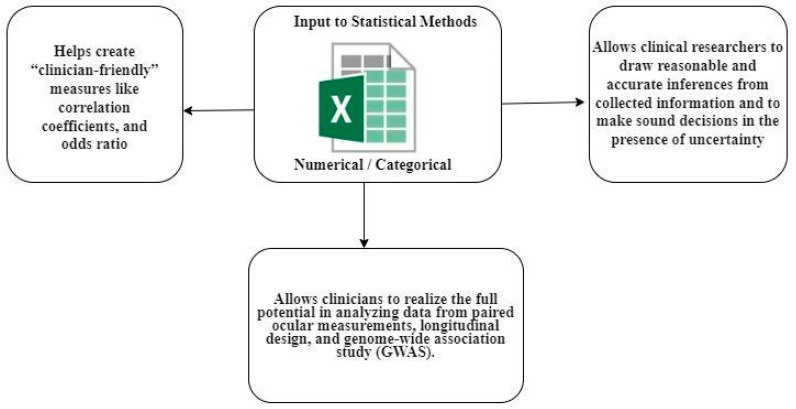
Application of statistical methods concerning the perspective of a clinician.

**Figure 4 diagnostics-13-00130-f004:**
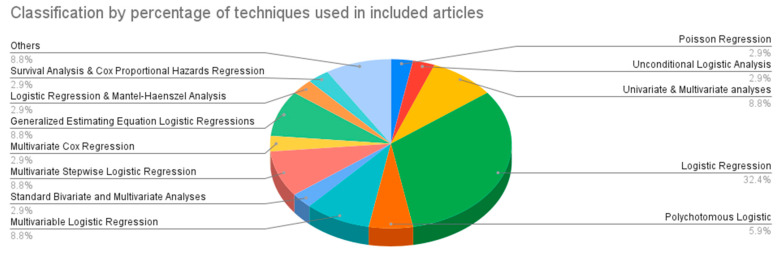
The classification by percentage of techniques used in included articles.

**Figure 5 diagnostics-13-00130-f005:**
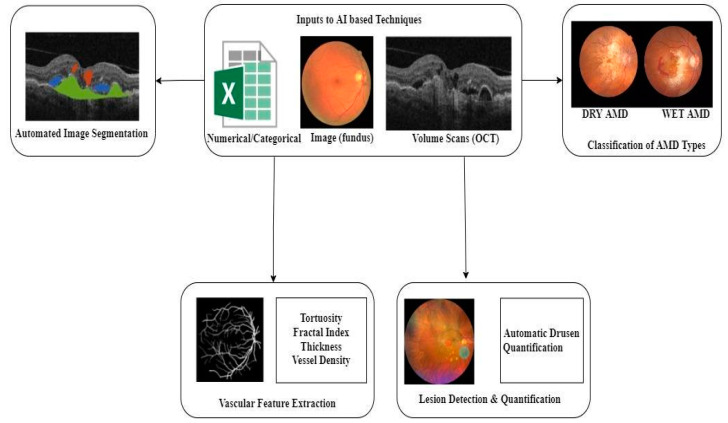
Applications of artificial intelligence from the perspective of a clinician.

**Table 1 diagnostics-13-00130-t001:** The summarized review of statistical techniques for AMD.

Technique	Source	Study Type	Country	Sex (%male)	Age Range (Years)	AMD Type	Classification Criteria	Adjustment	Risk Factors Identified
Poisson Regression Analysis	Holz et al. [29]	Prospective	London	46.8	Older than 50 years of age	AMD	Standardized grading scheme	Age, sex and smoking	Focal hyperpigmentation, slow choroidal filling and degree of confluence of drusen
Unconditional logistic analysis	Tamakoshi et al. [30]	Case-control	Japan	100	Aged 50 to 69 years	Neovascular AMD	NR	Age, sex	Cigarette smoking
Univariate and multivariate analyses	Klein et al. [31]	Population-based	United States	NR	43–86 years of age	ARM	WARMGS	Age and gender	(No strong relation between cardiovascular disease and most of its risk factors with the incidence of lesions associated with age-related maculopathy)
	Buch et al. [32]	Population-based cohort	Denmark	36.2	Between 60 and 80 years	ARM	Modification of WARMGS	Age and gender	Age, cataract, family history, alcohol consumption, the apolipoproteins A1 and B
	Women’s Health Initiative Sight Exam ancillary study [33]	Ancillary	United States	0	63 years and older.	Late AMD	WARMGS	Age	Smoking, use of calcium channel blockers, diabetes, and obesity
Logistic Regression	Chaine et al. [34]	Case-control	France	31	50–85 years	AMD	NR	NR	Arterial hypertension, coronary disease, hyperopia, light-coloured irises, lens opacities and previous cataract surgery
	POLA study [35]	Prospective	France	43.8	60 years or over	AMD	International classification *	Age and gender	(No significant association of late AMD with a history of cardiovascular disease, diabetes, and hypertension)
	Vine et al. [36]	Case-control	United States	41.8	≥65 Year old	AMD	NR	Age, CRP, and homocysteine level	CRP and homocysteine level
	AREDS study [37]	Clinic-based prospective cohort	NR	NR	55 to 80 years	Neovascular AMD	NR	Age, gender, and AREDS treatment	Smoking, race, and BMI
	Fraser-Bell et al. [38]	Population-based, cross-sectional	United States	42	40 years old	Early and advanced AMD	Modified WARMGS	Age, sex and smoking status	Smoking and heavy alcohol consumption
	Gemmy et al. [39]	Population-based, cross-sectional	Singapore and India	50.2 (Singapore) & 47.3 (India)	40–83 years	Early AMD	International classification of the Wisconsin age-related maculopathy	Age, BMI, sex, cholesterol, myocardial infarction, hypertension, central corneal thickness axial length, and IOP.	Shorter axial length higher BMI, previous cataract surgery, lower cholesterol and hypertension.
	Yip et al. [40]	Prospective cohort	United Kingdom	43.1	44–91 years	AMD	Modified Wisconsin protocol *	Sex, education, smoking, and SBP.	Older age, baseline CRP, and a higher baseline and follow-up levels of HDL.
	Raman et al. [41]	Population-based cross-sectional	India	NR	≥60 years	Early and late AMD	International ARM epidemiological study group	Age and gender	Age per year increase, middle socioeconomic status, and smokeless tobacco
	Myra et al. [42]	Observational	Australia	40	47–85 years	Late AMD	NR	Sex, age at fundus photography, index of relative socioeconomic disadvantage, and the Mediterranean diet score	Current smokers
	Connolly et al. [43]	Cohort	Ireland	44	≥50 years	AMD	A modified version of the international classification and grading system for AMD	Age, sex, education and CFH	Older age, the presence of ARMS2 and CFH risk alleles
	Butt et al. [44]	Cross-sectional	United States	NR	45 to 74 years	Early and late AMD	University of Wisconsin ocular epidemiology reading center	NR	Age and HDL cholesterol
Polychotomous logistic regression analyses	Hyman et al. [45]	Case-control	United States	40	Between the age of 50 and 79 years	Neovascular AMD	Independent graders at the reading center	Age, sex, and energy intake.	Moderate to severe hypertension
	AREDS study [46]	Case-control	United States	44.2	Aged 60 to 80 years	AMD	The Wisconsin age-related maculopathy grading system #	Age and gender	Smoking, hypertension, lens opacities, hyperopia, female gender, less education, white race, and increased BMI
Multivariable logistic regression models	Klein et al. [47]	Cohort	United States	45.6	Aged 21 to 84 years	AMD	WARMGS	Age and sex	Smoking and serum HDL cholesterol
	Shim et al. [48]	Prospective cohort	South Korea	60.5	Older than 50 years	Early AMD progression	International age-related maculopathy (ARM) epidemiological study group and WARMGS	Age, alcohol consumption, smoking status, BMI, BP, HDL cholesterol, and total cholesterol	An increasing number of drusen, central drusen location, hypertension, and current smoking.
	Erke et al. [49]	Population-based, cross-sectional	Norway	43	65–87 years	AMD and late AMD	International classification system *	Age, sex, smoking and SBP	Smoking, higher SBP, physical inactivity, overweight and obesity in women
Standard Bivariate and Multivariate Analyses	Krishnaiah et al. [50]	Population-based, cross-sectional	India	47	Aged 40 to 102 years	AMD	International classification and grading system	Age, area and gender	Ageing, smoking, prior cataract surgery, and presence of cortical cataract.
Multivariate stepwise logistic regression	Choudhury et al. [51]	Population-based prospective cohort	United States	39.1	Aged 40 or older	Any AMD and progression of AMD	Modified WARMGS	Age	Older age, current smoking and higher pulse pressure
	Jonasson et al. [52]	Population-based prospective cohort	Iceland	42.4	Aged 67 years and older	AMD	Modification of WARMGS	Age and sex	Age, smoking, plasma HDL cholesterol, BMI and female sex
	Saunier et al. [53]	Population-based cohort	France	37.3	73 years or older	Early to advanced AMD	International classification * and to a modification of the grading scheme used in the multi-ethnic study of atherosclerosis @	Age and sex	Fellow eye, smoking, plasma HDL cholesterol concentration, and CFH Y402H genotype
Multivariate Cox regression survival analysis	Lechanteur et al. [54]	Retrospective	Netherlands	34.3	54.3–93.4 years.	End-stage AMD	NR	Age, education, sex, baseline AMD grade, smoking, BMI, six genetic variants and associated genotypes, and treatment groups	Sex, smoking status, age, to a lesser extent BMI, CFI (rs10033900) and LPL (rs12678919)
Generalized estimating equation logistic regressions	Cougnard et al. [55]	Population-based	France	38.1	65 years and older	Early and any AMD	International classification *@	Age, educational level, sex, BMI, smoking, lipid-lowering medication use for all relevant genetic polymorphisms, cardiovascular disease and diabetes,	HDL, TC, LDL, higher HDL, and TG
	Foo et al. [56]	Population-based cohort	Singapore	49.7	NR	Early AMD	WARMGS	Age, gender, hypertension, total cholesterol, cardiovascular disease, BMI categories, smoking status, alcohol consumption frequency, serum CRP and ARMS2 genetic loci.	Heavy alcohol drinking, underweight BMI, ARMS2 rs3750847 homozygous genetic loci carrier, and cardiovascular disease history.
	Wang et al. [57]	Population-based cohort	Australia	39.2	49 years or older	AMD	WARMGS	Age, sex, smoking status and the correlation between eyes	Eyes with indistinct soft drusen, large drusen, hyperpigmentation or a large area of the macula covered by drusen.
Logistic regression analyses and Mantel-Haenszel analysis	Aoki et al. [58]	Cross-sectional	Japan	60	65–74 years and 75–84 years	AMD	Simplified severity scale for AMD from the AREDS	Age	CFH I62V and ARMS2 A69S variant
Survival analysis and Cox proportional hazards regression	Hallak et al. [59]	Retrospective, post hoc secondary analysis	United States	40.8	50 years or older	Neovascular AMD	NR	NR	Mean drusen reflectivity, the total en-face area of the drusen restricted to a circular area of 3 mm from the fovea and one genetic variant (rs61941274)
Others	Hammond et al. [60]	Case-control	United States	47	NR	Neovascular AMD	NR	NR	Smokers
	Alain et al. [61]	Case-control	France	22.6	Mean age 77 years	AMD	WARMGS	NR	Perturbations of HDL metabolism
	Tomany et al. [62]	Population-based cohort	Australia, Netherlands, and the United States	43	43–95 years	AMD	Wisconsin and international age-related maculopathy grading systems	Age, gender (when appropriate), data source, and follow-up time	Smoking

AMD, age-related macular degeneration; ARM, age-related maculopathy; WARMGS, the Wiscon-sin age-related maculopathy grading system; CRP, C-reactive protein; BMI, body mass index; POLA, Pathologies Oculaires Liées à I’Age; AREDS, Age-Related Eye Disease Study; IOP, intraocular pressure; SBP, systolic blood pressure; HDL, high-density lipoprotein; CFH, complement factor H; ARMS2, age-related maculopathy susceptibility 2; BP, blood pressure; SBP, systolic blood pressure; LPL, lipoprotein lipase; CFI, complement factor I; LDL, low-density lipoprotein; TC, total cholesterol; TG, triglycerides. *: Bird AC, Bressler NM, Bressler SB, Chisholm IH, Coscas G, Davis MD, et al. An international classification and grading system for age-related maculopathy and age-related macular degeneration. The international ARM epidemiological study group. Surv Ophthalmol. 1995;39:367–74. #: Klein R, Davis MD, Magli YL, et al. The Wisconsin age-related maculopathy grading system. Ophthalmology 1991;98:1128–34. @: Klein R, Klein BE, Knudtson MD, et al. Prevalence of age-related macular degeneration in 4 racial/ethnic groups in the Multi-Ethnic Study of Atherosclerosis. Ophthalmology. 2006;113(3): 373–380.

**Table 2 diagnostics-13-00130-t002:** The summarized review of AI techniques for AMD.

Source	Technique	Dataset	Metrics	Disease
Grinsven et al. [64]	Supervised machine learning algorithm	A total of 407 images of different eyes with nonadvanced stages of AMD (i.e., stages 1, 2, and 3 according to the criteria shown in Table 1), with sufficient grading quality for human graders, were selected consecutively from the European genetic database (EUGENDA), a large multicenter database for clinical and molecular analysis of AMD.	AUROC values of 0.948 and 0.954	AMD risk assessment
Grinsven et al. [65]	Machine learning algorithm	A set of subjects with and without RPD were selected from the Rotterdam Study. A prospective cohort study aimed to investigate risk factors for chronic diseases in the elderly.	AUROC value of 0.941	Reticular pseudo drusen (RPD) detection
Liefers et al. [66]	Deep learning model	This study’s imaging data (OCT B scans) were obtained from 30,337 patients at five centres in the UK (NRES Committee London, City Road and Hampstead, London).	On 11 of 13 features, the model obtained a mean Dice score of 0.63 ± 0.15, compared with 0.61 ± 0.17 for the observers. ICC was 0.66 ± 0.22, compared with 0.62 ± 0.21 for the observers	Feature segmentation associated with neovascular and atrophic AMD
Schmidt-Erfuth et al. [67]	Random forest regression algorithm	Data (spectral-domain (SD) OCT volume scans) of 614 evaluable patients receiving intravitreal ranibizumab monthly or pro re nata according to protocol-specified criteria in the HARBOR trial were studied.	At baseline, OCT features and BCVA were correlated with R^2^ = 0.21.	Predict best-corrected visual acuity (BCVA) outcomes
Schmidt-Erfuth et al. [68]	Deep learning method (convolutional neural network (CNN))	SD-OCT scans of 1095 patients enrolled in the HARBOR trial	NR	Measure fluid response to anti-vascular endothelial growth factor (VEGF) treatment in neovascular AMD
Keenan et al. [69]	Artificial Intelligence Algorithms	Data from (a) the HARBOR trial, (b) a tertiary referral retinal centre in the United Kingdom, (c) a tertiary referral retinal centre in Israel, and (d) the AREDS2 10-year follow-up. were studied,	Large ranges that differed by population were observed at the treatment-naive stage: 0–3435 nL (IRF), 0–5018 nL (SRF), and 0–10,022 nL (PED).	Validation of retinal fluid volumes (IRF, SRF and PED)
Lee et al. [70]	Automated segmentation algorithm with a CNN	A dataset including 930 B-scans from 93 eyes of 93 patients with nAMD.	Dice coefficients for segmentation of IRF, SRF, SHRM, and PED were 0.78, 0.82, 0.75, and 0.80	To quantify and detect intraretinal fluid (IRF), subretinal fluid (SRF), pigment epithelial detachment (PED), and subretinal hyperreflective material (SHRM) with nAMD
Yim et al. [71]	Artificial intelligence system	A cohort of 2,795 patients (OCT scans) across seven different sites who were first diagnosed with nAMD between June 2012 and June 2017	Sensitivity of 80% at 55% specificity and 34% specificity at 90% sensitivity	Progression to exudative wet AMD
Yan et al. [72]	Modified deep convolutional neural network	The data consisted of 52 AMD-associated genetic variants and 31,262 fundus images from 1,351 subjects from the age-related eye disease study (AREDS) fundus images coupled with genotypes.	AUC value of 0.85	AMD progression
Peng et al. [73]	Deep learning (DL) and survival analysis	AREDS and AREDS2	5-year C-statistic 86.4	Late AMD
Ajana et al. [74]	Prediction model used bootstrap lasso for survival analysis	The training data set included Rotterdam study I (RS-I) enrolled participants.	AUC estimation in RS-I was 0.92 at five years, 0.92 at ten years and 0.91 at 15 years	Advanced AMD
Seddon et al. [77]	Predictive model	The data was from 1446 individuals who participated in the multicenter AREDS, of which 279 progressed to advanced AMD and 1167 did not progress during 6.3 years of follow-up	C statistic score of 0.83	Prevalence and incidence of AMD
Seddon et al. [79]	Model of AMD progression	Data consisted of 2937 individuals in the AREDS	AUC 0.915 in the total sample	AMD Progression
Klein et al. [80]	Risk assessment model	Longitudinal data from 2846 participants in the AREDS	C statistic = 0.872. Brier score at 5 years = 0.08	Advanced AMD
Seddon et al. [81]	Predictive model and online application	Data from the AREDS for Caucasian participants were used for this analysis	AUC- 91.1	Progression to advanced AMD
Spencer et al. [82]	Logistic regression and grammatical evolution of neural networks (GENN) models	A VM family dataset, the population-based age-related maculopathy ancillary (ARMA) study cohort, and Vanderbilt-Miami (VM) clinic-based case-control dataset.	Sensitivity of 77.0% and specificity of 74.1%	High- and low-risk groups for AMD
Fraccaro et al. [83]	Random forests, AdaBoost and SVM, as well as white-box methods, including decision trees and logistic regression	Data on healthy subjects, study participants, and patients with macular diseases were collected from March 2013 to January 2014 during routine clinical practice at the Medical Retina Center of the University Eye Clinic of Genoa (Italy).	Logistic Regression, AdaBoost, and random forests achieved a mean AUC of 0.92, followed by decision trees and SVM with a mean AUC of 0.90.	Diagnose AMD
Shin et al. [84]	Risk prediction model	The study sample included 50 years of age or older individuals counting 10,890; 318 (2.92%) presented with early AMD findings in baseline examinations.	C statistic-0.84	Progression of AMD

## Data Availability

Datasets available on request form corresponding author only as the data are sensitive and participants may be potentially identifiable.

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
