# Peer review of "The Need for Artificial Intelligence Based Risk Factor Analysis for Age-Related Macular Degeneration: A Review"

_diagnostics, 2022, doi:10.3390/diagnostics13010130_

Round 1
Reviewer 1 Report
The authors made some reasonable efforts to write a comprehensive review on Artificial Intelligence based Risk Factor Analysis for age-related macular degenerations, there are some serious concerns with this study:
Abstract Section:
1. It is recommended to add a comparative analysis of published review materials.
2. Add a motivation sentence to distinguish your work from the rest of the studies.
Introduction Section:
3. It is recommended to add a brief description of eye diseases like cataract [1-4], glaucoma [5] and diabetic retinopathy [6-7] and then starts from AMD.
[1]. https://scholar.google.com/citations?view_op=view_citation&hl=en&user=VJv0M4cAAAAJ&citation_for_view=VJv0M4cAAAAJ:4DMP91E08xMC
[2]. https://scholar.google.com/citations?view_op=view_citation&hl=en&user=VJv0M4cAAAAJ&citation_for_view=VJv0M4cAAAAJ:4TOpqqG69KYC
[3]. https://scholar.google.com/citations?view_op=view_citation&hl=en&user=VJv0M4cAAAAJ&citation_for_view=VJv0M4cAAAAJ:5nxA0vEk-isC
[4]. https://scholar.google.com/citations?view_op=view_citation&hl=en&user=VJv0M4cAAAAJ&citation_for_view=VJv0M4cAAAAJ:8k81kl-MbHgC
[5]. https://scholar.google.com/citations?view_op=view_citation&hl=en&user=VJv0M4cAAAAJ&cstart=20&pagesize=80&citation_for_view=VJv0M4cAAAAJ:R3hNpaxXUhUC
[6]. https://scholar.google.com/citations?view_op=view_citation&hl=en&user=VJv0M4cAAAAJ&citation_for_view=VJv0M4cAAAAJ:L8Ckcad2t8MC
[7]. https://scholar.google.com/citations?view_op=view_citation&hl=en&user=VJv0M4cAAAAJ&cstart=20&pagesize=80&citation_for_view=VJv0M4cAAAAJ:e5wmG9Sq2KIC
4. The number of people with AMD is predicted to increase from 196 million in 2020 to 288 million by 2040 [3]. Rather than simply citing the work, it is recommended to add the current W.H.O. report on vision statistics.
5. Add paper organization at the end of the introduction section.
Methods Section:
6. Why the recent articles of 2020-2022 are not considered in this study?
Artificial Intelligence in AMD Section:
7. It is recommended to add the table in section 4.1 and 4.2, as mentioned in 4.3
Significance of AI over traditional Statistical Methods Section:
8. Add the subheading open problems and indicate the problems with existing techniques followed by some solutions as future research directions.
Conclusion Section:
9. It can be improved by adding comparative analysis with published relevant reviews. Also, there should be a description of the best statistical results achieved using any techniques to give proper insight to the readers.
Author Response
Dear Reviewer 1 ,
Thank you very much for your timely processing of our submission and your useful feedback. We would also like to thank you for your timely processing and valuable comments which have helped us a lot to improve the quality of the manuscript.
Following the decision for major revision, we have incorporated the changes suggested by you . Our detailed point-to-point response to the reviewers’ comments is provided in the attachment.
Thanks,
Rajiv Raman

Reviewer 2 Report
In this manuscript a review of AI applications for the determination of the risk factor for macular edema is presented.
The review is well written and focused. However,
1) a significant improvement consists in discussing which are the best models also considering the costs of obtaining the dataset in relation to the accuracy(or other validation metrics) of AI models. For example, suppose that a method using an expensive acquisition method has a prediction accuracy that is only moderately higher than a less accurate but cheaper method.
2)Which are the future innovations? Novel analysis methods? novel biomarkers? They could be mutuated from related ocular pathologies: for example you can refer, as regards biomarkers, to angiogenesis related markers in rethinopathy (https://www.ncbi.nlm.nih.gov/pmc/articles/PMC5397989/) or to novel bioimaging markers such as fluidity of red blood cells (https://pubmed.ncbi.nlm.nih.gov/33210748/). For what concerns analysis methods, novel segmentation methods to unveil metabolic features (https://pubmed.ncbi.nlm.nih.gov/33516373/ and https://pubmed.ncbi.nlm.nih.gov/32248131/)
Author Response
Dear Reviewer 2,
Thank you very much for your timely processing of our submission and your useful feedback. We would also like to thank you for your timely processing and valuable comments which have helped us a lot to improve the quality of the manuscript.
Following the decision for major revision, we have incorporated the changes suggested by you. Our detailed point-to-point response to the reviewers’ comments is provided in the attachment.
Thanks,
Rajiv Raman

Round 2
Reviewer 1 Report
The authors have well addressed the suggested comments, and now this article is considered for publication.